# RBF Neural Network Based Backstepping Control for an Electrohydraulic Elastic Manipulator

**Duc-Thien Tran**, **Minh-Nhat Nguyen** and **Kyoung Kwan Ahn** *

School of Mechanical Engineering, University of Ulsan Daehakro 93, Nam-gu, Ulsan 680-764, Korea;
thientd@hcmute.edu.vn (D.-T.T.); nm.nhat@hutech.edu.vn (M.-N.N.)
* Correspondence: kkahn@ulsan.ac.kr; Tel.: +82-52-259-2282

**Abstract:** An electrohydraulic elastic manipulator (EEM) is a kind of variable stiffness system (VSS). The equilibrium position and stiffness controller are the two main problems which must be considered in the VSS. When the system stiffness is changed for a specific application, the system dynamics are significantly altered, which is a challenge in controlling equilibrium position. This paper presents adaptive robust control for controlling the equilibrium position of the EEM under the presence of the variable stiffness. The proposed control includes sliding mode controls (SMCs), radial basis function neural network (RBFNN), and backstepping technique. The RBFNN is employed to compensate for the uncertainties and the variant stiffness in mechanical dynamics and hydraulic dynamics. The Lyapunov approach and projection algorithm are used to derive the adaptive laws of the RBFNN and to prove the stability and robustness of the entire EEM. Finally, some experiments are implemented and compared with other controllers to prove the effectiveness of the proposed method with the variant stiffness.

**Keywords:** electrohydraulic system; sliding mode control; backstepping control; variable stiffness; projection algorithm; RBF neural network

## 1. Introduction

Nowadays, many researchers have focused on developing high-performance machines with capabilities comparable to humans, especially with respect to motion, safety, as well as energy efficiency. From the analysis of human and animal behaviors, it was found that the adaptable compliance and variable stiffness play important roles. Recently, actuators and systems have been developed with the ability to adjust the stiffness, called a variable stiffness series elastic system (VSSES). It is a kind of variable stiffness system, which consists of two actuators where one regulates the equilibrium position of the VSSES, and the other adjusts its stiffness. On an aspect of safety of the robot manipulator, a series elastic manipulator (SEM) contains a variable stiffness actuator (VSA) providing the ability to decouple the inertia of the actuator proper from the inertia of the last link. That makes robots safe and improves their adaptability in the human–robot cooperation field [1]. In literature, many applications of the variable stiffness system (VSS) [2] such as shock absorbance [3–5], stiffness variability with a constant load [6,7], and cyclic movement [8], the safety robot [9] proved the effectiveness of the variable stiffness system although this research focused on tuning the system stiffness optimally. Normally, primary torque or force of these VSSs is generated by electric motors [3–5,8,9], or pneumatic actuators [6,7,9]. Therefore, it is difficult to use these VSSs for applications which require high power and compact size. To enhance the VSS working range with the above abilities at high stiffness states, an electrohydraulic actuator is a good choice to develop a new VSS because of its advantages of small size-to-power ratios and large force/torque output [10]. However, high nonlinearity and uncertainties in its dynamics

can increase the complexity of the system dynamics, which are great challenges associated with the control VSS.

To control a nonlinear system under the presence of uncertain environments and external disturbances, sliding mode control (SMC), a nonlinear robust control, is utilized [11]. The SMC strategies [12–14] have been effectively developed in many systems with matched uncertainties which mean the uncertainties and the control input act on exactly the same channel [15]. However, its main drawback is a chattering effect, which can affect the control performance [16]. In order to solve this issue and enhance the control performance, the SMC has been combined with some adaptive strategies, such as an integral adaptive switching gain, adaptive parametric uncertainties, and adaptive neural network. The integral adaptive laws [17] did not require any knowledge of the uncertainties. The robust gains increase until they bound the maximum uncertainties, then the robustness and stability are guaranteed. However, when the magnitude of the uncertainties is significantly smaller than the robust gain, the chattering issue can still occur. The adaptive laws [18,19] were derived to estimate the parametric uncertainties in a hydraulic system. The quality of these adaptive laws depends on the known dynamical model. Sliding mode controls [20–23] with radial basis function neural networks (RBFNNs) were used to approximate the unknown nonlinearities and the upper bound of the estimated disturbances of a three-phase shunt active power filter by simulations. Although these approximations did not require any knowledge of the uncertainties, their adaptive laws are sensitive to external disturbances and measurement noise [24].

Motivated by the previous works about the variable stiffness system and SMC, an electrohydraulic elastic manipulator (EEM) which includes an adjustment based stiffness mechanism (ABSM) and an electrohydraulic system (EHS) has been investigated as a VSSES. This paper proposes an adaptive robust control for controlling the equilibrium position of the EEM with the presence of the variant stiffness and the uncertainties. In order to improve the control performance, the actuator dynamics are included in the system dynamics, which gives rise to the matched and unmatched uncertainties. The backstepping approach is well-known as a strong method to handle these uncertainties which have low dynamics [25]. Then, the proposed control is developed based on the SMCs and RBFNN with the backstepping technique to use their advantages. The main contributions of this paper are presented as follows:

(1) The RBFNN is provided to approximate the uncertainties in both the hydraulic and mechanical dynamics of the EEM system. Especially, the learning laws of the weighting vector in the RBFNNs are produced by the Lyapunov approach and the project algorithm to improve the robustness of the learning laws.

(2) The Lyapunov approach and the backstepping technique are used to prove the stability of the control system with the presence of matched and unmatched uncertainties.

(3) Some experiments are implemented on the real test rig, and the results are compared to the Proportional integral derivative (PID) control and the conventional backstepping sliding mode control (CBSMC) to verify the effectiveness of the proposed control with the variant stiffness.

The remainder of this paper is organized as follows: Section 2 presents the electrohydraulic elastic manipulator dynamics. Next, conventional backstepping control and the proposed control are provided in Section 3. In Section 4, the experimental results are presented. Conclusions are shown in Section 5. The appendixes present the definitions of the matrices, vectors, and functions.

## 2. Electrohydraulic Elastic Manipulator Dynamics

The proposed EEM is presented in Figure 1. An ASBM system connects between the first and second link of the EEM. The first link of the EEM is driven by an EHS which contains a hydraulic power unit, a single-rod hydraulic cylinder, and two pressure sensors. The cylinder rod is connected to the second rotary shaft and controls the equilibrium position of the manipulator. The ASBM includes two springs, ball screws, and an electric motor. The stiffness is obtained by regulating the tension of

the springs. Extending both springs makes the outer link stiffer while relaxing both springs makes the outer link more compliant. The springs are mounted to the ball screw nut, and the ball screw is driven by an electric motor, so the springs are either extended or relaxed by controlling the position of the motor. The safety and minimized energy consumption of the EEM system will be obtained when the positioning strategy of the motor is optimally generated.

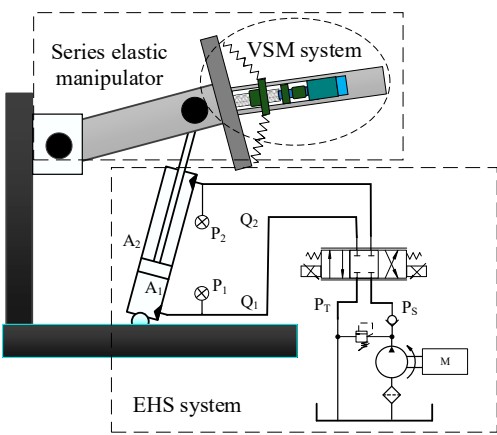

**Figure 1.** Electrohydraulic elastic manipulator.

### 2.1. Dynamics of the Electrohydraulic Servo System

The EHSs consists of a servo-valve and a linear hydraulic actuator, which is powered by a gear pump as shown in Figure 1. The relationship between the input and the spool displacement is presented as follows:

$$x_s = k_s u \tag{1}$$

where $x_s$ is the spool displacement of the servo-valve (m); $k_s$ is the spool displacement gain (m/V); and $u$ is the input voltage of the servo-valve (V).

**Remark 1.** *In many EHSs, valve dynamics are faster than the dynamics of the other parts of the system. Additionally, the servo-valves are usually controlled by valve drivers. So, valve dynamics can be ignored without significant sacrifice of control performance.*

The actuator dynamics can be described as [26]

$$\frac{V_1}{\beta} \dot{P}_1 = -A_1 \dot{x}_c - C_{\text{leak}}(P_1 - P_2) + Q_1 \tag{2}$$

$$\frac{V_2}{\beta} \dot{P}_2 = A_2 \dot{x}_c + C_{\text{leak}}(P_1 - P_2) - Q_2 \tag{3}$$

where $V_i = V_{0i} + (-1)^{i-1} A_i x_c, (i = 1, 2)$ is total control volume of the $i$th chamber; $x_c$ is displacement of the cylinder; $V_{0i}(i = 1, 2)$ are initial volume of two chambers; $A_1$ and $A_2$ are the rod area of the cylinder and the head-side area of the cylinder, respectively; $\beta$ is effective bulk modulus; $C_{\text{leak}}$ is coefficient of the internal leakage; $Q_1$ is supplied flow rate to the forward chamber; and $Q_2$ is return flow rate to the return chamber. $Q_i(i = 1, 2)$ are related to the spool displacement of the servo-valve $x_s$ [26].

$$Q_1 = k_q \omega x_s \sqrt{\Delta P_1}, \quad \Delta P_1 = \begin{cases} (P_S - P_1), & x_s \geq 0 \\ (P_1 - P_T), & x_s < 0 \end{cases} \tag{4}$$

$$Q_2 = k_q \omega x_s \sqrt{\Delta P_2}, \quad \Delta P_2 = \begin{cases} (P_2 - P_T), & x_s \geq 0 \\ (P_S - P_2), & x_s < 0 \end{cases} \tag{5}$$

where $k_q$ is flow gain coefficient of the servo valve; $\omega$ is the servo valve area gradient; $P_S$ is supply pressure of the EHSs; $P_T$ is the tank pressure; and $P_1$ and $P_2$ are the pressure in the forward and return chambers, respectively.

In the cylinder actuator, the force can be calculated as follows:

$$F_c = A_1 P_1 - A_2 P_2 \tag{6}$$

### 2.2. Mechanical Dynamics

The two-link manipulator of the EEM is shown in Figure 2 with two different coordinate systems for the two degrees of freedom (DOF) of the manipulator. The parameters are also shown. $l_0$ presents the position of the 1st link along with the z-axis. $l_i (I = 1, 2)$ are the lengths of the links, and $\theta_i (i = 1, 2)$ are the angle of the *i*th from the coordinate system.

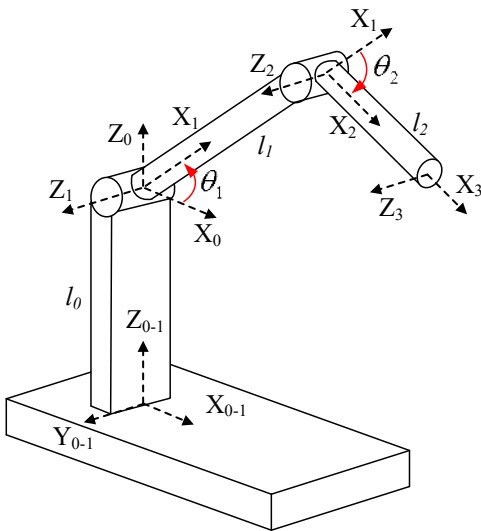

**Figure 2.** The architecture of the two-link robot manipulator in the electrohydraulic elastic manipulator (EEM).

The EEM dynamics without considering the adjustable stiffness mechanism are expressed based on the dynamics of a 2-DOF manipulator [27]. The dynamic equations are as follows:

$$M(\theta)\ddot{\theta} + C(\theta, \dot{\theta}) + G(\theta) = \tau + \tau_f \tag{7}$$

where $\dot{\theta}$, and $\ddot{\theta} \in R^{2\times1}$ are the joint velocity and acceleration vectors, respectively; $M(\theta) \in R^{2\times2}$ is the inertia matrix; $C(\theta, \dot{\theta}) \in R^{2\times2}$ presents the matrix of centripetal and Coriolis forces; $G(\theta) \in R^{2\times1}$ is the gravity vector; $\tau_f$ is the friction vector; and $\tau$ is the torque vectors. Detailed descriptions are given in Appendix A.

Figure 3 is a simple form of the EEM which is used to analyze the forces acting on the two-link manipulator: the force of hydraulic cylinder and the spring forces of the ASBM system. Three black lines represent the cylinder, the first, and the second link. From this figure, the excited torque produced by the hydraulic actuator can be obtained as

$$\tau_c = l_1 F_c \sin\gamma \tag{8}$$

where $\gamma$ is the angle between the first link and the cylinder. During the system operation, the angle $\gamma$ is around 90 degrees. So Equation (8) can be rewritten as follows:

$$\tau_c = l_1 F_c \tag{9}$$

Additionally, the relationship between the cylinder displacement $x_c$ and $\theta_1$ is expressed as:

$$\cos(\theta_0 + \theta_1) = \frac{l_1^2 + l_0^2 + d_0^2 - d_3{}^2}{2l_1\sqrt{\left(l_0^2 + d_0^2\right)}}$$

(10)

where $d_3 = d_{30} + x_c$, and $d_{30}$ is the length of the cylinder when $x_c = 0$.

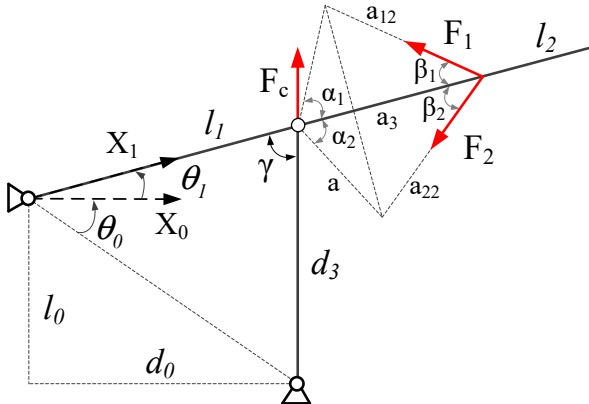

**Figure 3.** The forces acting on the two-link manipulator.

The angular motion of the second link is caused by the spring forces. The ASBM system dynamics are analyzed using the distance between the second rotary shaft and the spring position of $a_3$, the spring axes, and the geometric relationships of the VSA structure as shown in Figure 3.

The forces produced by the two springs are represented as follows:

$$\begin{aligned} F_1 &= \Delta x_1 K_1 \\ F_2 &= \Delta x_2 K_2 \end{aligned}$$

(11)

where $\Delta x_i (i = 1, 2)$ are the variations of the springs, and $K_i (i = 1, 2)$ are the stiffnesses of the springs.

These forces excite the coupling torques in the two links. The torques are expressed as follows:

$$\begin{aligned} \tau_{12} &= (F_1 \sin \beta_1 - F_2 \sin \beta_2) a_3 \\ \tau_{21} &= -(F_1 \sin \beta_1 - F_2 \sin \beta_2)(l_1 + a_3) \end{aligned}$$

(12)

where $\tau_{12}$ and $\tau_{21}$ present the coupling torques, which impact the two links and are caused by the springs in the adjustable stiffness mechanism, and $\beta_i$ is an angle between the ball screws and the $i$th spring.

The considered hydraulic friction torque in this model is a combination of the Coulomb friction and the viscous friction [28] as

$$\tau_{fc} = l_1 F_f = l_1\left(F_{Coul}\text{sign}\left(\dot{x}_c\right) + F_v \dot{x}_c\right)$$

(13)

where $F_{Coul}$ derives a Coulomb coefficient, and $F_v$ depicts a viscous coefficient.

The dynamics of the EEM can be represented as follows:

$$\ddot{\theta}_2 + f_2(\theta_2, \dot{\theta}_2) + g_2(\theta_2, \dot{\theta}_2)(\tau_{ext_2} + \tau_{12}) = 0$$

(14)

$$\ddot{\theta}_1 + f_1(\theta_1, \dot{\theta}_1) + g_1(\theta_1, \dot{\theta}_1)\left(\tau_{21} + \tau_{ext_1} + \tau_{fc} - \tau_{ehs}\right) = 0$$

(15)

where (14) is the outer link equation and $f_2(\theta_2, \dot{\theta}_2)$, and $g_2(\theta_2, \dot{\theta}_2)$ are known functions in the outer link dynamics; $\tau_{ext_2}$ is the external torques and the unknown function; and $\tau_{12}$ is coupling torque generated

by the variable stiffness. The EEM dynamic equation is represented by (15). $f_1(\theta_1, \dot{\theta}_1)$ and $g_1(\theta_1, \dot{\theta}_1)$ are the known functions of the manipulator dynamics; $\tau_{ext_1}$ is the external torques and modeling error.

When the system stiffness is adjusted by ASBM online, the coupling torque impacting the EHSs will alter with respect to the system stiffness. This issue and other uncertainties in the system are challenges for the position control of the EHSs. This paper proposes an indirect adaptive controller to guarantee the stability and robustness of the EHSs in the existence of uncertainties.

From (1), (4)–(6), and (15), the system dynamics are represented as follows:

$$
\begin{aligned}
\dot{x}_1 &= x_2 \\
\dot{x}_2 &= g_1 x_3 + f_1 + \delta_1(t) \\
\dot{x}_3 &= g_0 u + f_0 + \delta_2(t)
\end{aligned}
\tag{16}
$$

where $x = \begin{bmatrix} x_1 & x_2 & x_3 \end{bmatrix}^T = \begin{bmatrix} \theta_1 & \dot{\theta}_1 & F_{ehs} \end{bmatrix}^T$; $\delta_1(t)$ is the uncertainties, such as fiction, external torques, and modeling error, in the mechanical dynamics of the EEM; $\delta_2(t)$ is the uncertainties, such as leakages and modeling error, in the hydraulic dynamics; and $f_i (i = 0, 1)$ and $g_i (i = 0, 1)$ are defined in Appendix B.

**Assumption 1.** *The perturbation, $\delta_i(t)$, (i = 1, 2) changes with respect to time and is bounded by $\|\delta_i(t)\| \le \eta_i$.*

## 3. Control Design

In this section, two control design procedures are described. The first one is the robust CBSMC design [25] which has two control loops. An integral SMC is used to assure that the torque of the EHSs tracks a virtual torque. An SMC produces the virtual torque based on the feedback position and the desired position. The second one is the proposed control which is designed via the CBSMC and RBFNN approximation. The CBSMC is used to guarantee the robustness and stability of the controlled system, and RBFNN approximation is used to improve the precision of the control performance with the presence of the uncertainties. The proposed control structure is shown in Figure 4.

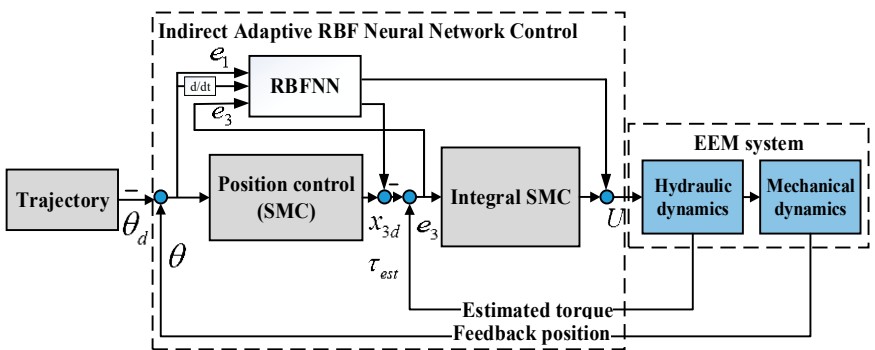

**Figure 4.** Structure of the adaptive backstepping sliding mode control.

### 3.1. Backstepping Sliding Mode Control

**Step 1:** The SMC is developed to make the tracking position error as small as possible. This step produces the virtual reference $x_{3d}$. The sliding surface is chosen as:

$$
s_1 = \lambda_1 e_1 + e_2
\tag{17}
$$

where $\lambda_1$ is a non-zero positive constant, $e_i = x_i - x_{id}, (i = 1, 2)$;

The derivative of the sliding variable is calculated as follows:

$$
\dot{s}_1 = \lambda_1 \dot{e}_1 + \dot{e}_2 = \lambda_1 \dot{e}_1 + g_1 x_3 + f_1 + \delta_1(t) - \dot{x}_{2d}
\tag{18}
$$

The virtual reference $x_{3d}$ is selected as:

$$x_{3d} = g_1^{-1}\left(-\lambda_1 \dot{e}_1 + \dot{x}_{2d} - f_1 - k_1 s_1 - \eta_1 sign(s_1)\right) \tag{19}$$

where $k_1$ is an arbitrary positive constant, and $\eta_1$ presents a robust gain of the SMC, $s_1$.

The different torque error is defined as follows:

$$e_3 = x_3 - x_{3d} \tag{20}$$

Considering the Lyapunov function candidate

$$V_1 = \frac{1}{2}s_1^2 \tag{21}$$

The derivative of the Lyapunov function (21) with (18) is shown as follows:

$$\dot{V}_1 = s_1 \dot{s}_1 = s_1\left(c_1 \dot{e}_1 + g_1 x_3 + f_1 + \delta_1(t) - \dot{x}_{2d}\right) \tag{22}$$

Applying the virtual reference $x_{3d}$ and (20) to the derivative of the Lyapunov function is represented as follows:

$$\dot{V}_1 = s_1\left(\lambda_1 \dot{e}_1 + g_1(x_{3d} + e_3) + f_1 + \delta_1(t) - \dot{x}_{2d}\right) = -k_1 s_1^2 + (\delta_1(t) - \eta_1 sign(s_1))s_1 + g_1 s_1 e_3 \tag{23}$$

**Step 2:** The controller is designed to guarantee that $e_3$ approaches zero. The integral sliding surface is chosen as follows:

$$s_2 = e_3 + \lambda_2 z_3 \tag{24}$$

where $\lambda_2$ is a positive constant, and $z_3 = \int_0^t e_3(\tau)d\tau$.

Taking the derivative of the sliding variable (24), its result is shown as follows

$$\dot{s}_2 = g_0 u + f_0 + \delta_2(t) - \dot{x}_{3d} + \lambda_2 e_3 \tag{25}$$

Since the virtual reference in (19) includes a discontinuous term, the derivative of the sliding variable in (25) cannot be expressed. In order to deal with this problem, a tan hyperbolic function replaced the discontinuous term. The virtual control (19) is represented as follows:

$$x_{3d} = g_1^{-1}(-\lambda_1 \dot{e}_1 + \dot{x}_{2d} - f_1 - k_1 s_1 - \eta_1 \tanh(\frac{s_1}{\phi_1})) \tag{26}$$

The control is chosen as:

$$u = g_0^{-1}(\dot{x}_{3d} - f_0 - k_2 s_2 - g_1 s_1 - \lambda_2 e_3 - \eta_2 \tanh(\frac{s_2}{\phi_2})) \tag{27}$$

where $\phi_{i(i=1,2)}$ are arbitrary positive constants, $k_2$ is arbitrary positive constants, and $\eta_2$ is a robust gain of the SMC, $s_2$.

Considering the Lyapunov function candidate:

$$V_2 = V_1 + \frac{1}{2}s_2^2 = \zeta^T \Omega \zeta \tag{28}$$

where $\zeta = [s_1, e_3, z_3]^T$, and $\Omega = \frac{1}{2}\begin{bmatrix} 1 & 0 & 0 \\ 0 & 1 & \lambda_2 \\ 0 & \lambda_2 & \lambda_2^2 \end{bmatrix}$.

The time derivative of the Lyapunov function (28) is computed as follows:

$$\dot{V}_2 = \dot{V}_1 + s_2 \dot{s}_2 \tag{29}$$

When (25) and (27) are used in (29), the derivative can be rewritten as follows:

$$
\begin{aligned}
\dot{V}_2 \;=& -k_1 s_1^2 + s_1\left(\delta_1(t) - \eta_1 \tanh\left(\tfrac{s_1}{\phi_1}\right)\right) + g_1 s_1 e_3 + s_2\left(g_0 u + f_0 + \delta_2(t) - \dot{x}_{3d} + \lambda_2 e_3\right) \\
=& -k_1 s_1^2 + s_1\left(\delta_1(t) - \eta_1 \tanh\left(\tfrac{s_1}{\phi_1}\right)\right) + g_1 s_1 e_3 - k_2 s_2^2 + s_2\left(\delta_2(t) - \eta_2 \tanh\left(\tfrac{s_2}{\phi_2}\right)\right) - g_1 s_1 s_2 \\
=& -\sum_{i=1}^{2}\left(k_i s_i^2 + s_i\left(\delta_i(t) - \eta_i \, sign\left(\tfrac{s_i}{\phi_i}\right)\right) + s_i \eta_i\left(sign\left(\tfrac{s_i}{\phi_i}\right) - \tanh\left(\tfrac{s_1}{\phi_1}\right)\right)\right) - \lambda_2 g_1 s_1 z_3 \\
\le& -\sum_{i=1}^{2}\left(k_i s_i^2 + (\delta_i(t) - \eta_i)|s_i|\right) - \lambda_2 g_1 s_1 z_3 + \varepsilon_d \le -\zeta^T \Xi \zeta + \varepsilon_d \\
\le& -\lambda_{\min}\left(\Xi \Omega^{-1}\right) V_2 + \varepsilon_d
\end{aligned}
\tag{30}
$$

where $\Xi = \begin{bmatrix} k_1 & 0 & \tfrac{\lambda_2 g_1}{2} \\ 0 & k_2 & k_2 \lambda_2 \\ \tfrac{\lambda_2 g_1}{2} & k_2 \lambda_2 & k_2 {\lambda_2}^2 \end{bmatrix}$, $\left\| \sum\limits_{i=1}^{2} s_i \eta_i\left(sign(s_i) - \tanh\left(\tfrac{s_i}{\phi_i}\right)\right)\right\|_{\infty} \le \varepsilon_d$ is a positive constant, and $\lambda_{\min}((.))$ is the minimum eigenvalue of matrix $(.)$.

When Assumption 1 is satisfied, and the parameters $k_1$, $k_2$, and $\lambda_2$ are chosen so the matrix $\Xi$ is a semi-positive definite matrix, the controlled system will be uniformly asymptotically stable [29].

In practice, the boundaries of the uncertainties, $\delta_i (i = 1, 2)$, are difficult to determine. If the robust gains chosen are smaller than the boundaries of the uncertainties, then the stability and robustness are not assured. In another way, if the robust gains selected are significantly larger than the boundaries, then the stability and robustness are guaranteed, but the chattering effects can happen and affect the control performance.

*3.2. Adaptive Backstepping Control Based RBFNN*

In order to handle this issue and improve the precision position, the adaptive backstepping control based RBFNN is employed with the structure in Figure 4. The proposed control developed an adaptive approximation via the radial basis function neural network [30]. These approximations are used to compensate for the uncertainties in the mechanical and hydraulic dynamics.

3.2.1. Adaptive Approximation via RBFNN

As shown in Figure 5, the RBFNN with four layers, which are the input layer, selection layer, hidden layer, and output layer, is employed to implement the approximations. The inputs and the output of the RBFNN are the tracking errors and the control input, respectively. The functions of each layer are presented as follows:

The input layer rescaled the input variables, $e_i (i = 1, \ldots, p)$ to the next layers.

The selection layer chooses the input for each approximation, $v_{ij} \in R^{p \times 1} (i = 1, 2; j = 1 \ldots m)$.

$$\gamma_{ij} = v_{ij}^T E, \, (i = 1, 2; j = 1, \ldots, m) \tag{31}$$

with $\sum\limits_{k=1}^{p} v_{ijk} = 1$, $\|v_{ij}\|_2 = 1$, and $E = \left[e_1, \ldots, e_p\right]^T \in R^{p \times 1}$.

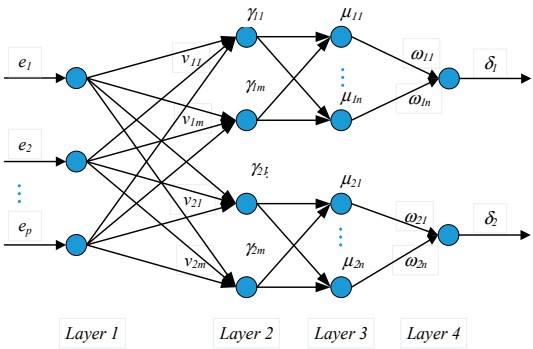

**Figure 5.** Structure of the radial basis function neural network (RBFNN).

The hidden layer derives the input values with the radius basis function, Gaussian function, as follows:

$$\mu_{ij} = \exp\left(\frac{-\left(G_i - m_{eij}\right)^T \left(G_i - m_{eij}\right)}{\sigma_{ij}^2}\right), (i = 1, 2; j = 1, \ldots, n) \tag{32}$$

where $G_i = \begin{bmatrix} \gamma_{i1} & \cdots & \gamma_{im} \end{bmatrix}^T \in R^{m\times1}$ is the input vector, and $m_{eij} \in R^{m\times1}$ and $\sigma_{ij}(i = 1, 2; j = 1, \ldots, n)$, respectively, are the mean vector and the standard derivation of the Gaussian functions of the node $ij$ in the hidden layer.

The output layer presents the compensation signals for the mechanical dynamics and the hydraulic dynamics. Each node $\delta_i(i = 1, 2)$, which is calculated as follows:

$$\delta_i = \sum_{j=1}^{m} w_{ij}\mu_{ij}(G_i) \tag{33}$$

These equations can be represented in the vector form as follows:

$$D = \begin{bmatrix} \delta_1 & \delta_2 \end{bmatrix}^T = trace\left(W^T\mu\right) \tag{34}$$

where $W = \begin{bmatrix} W_1 \\ W_2 \end{bmatrix} = \begin{bmatrix} w_{11} & \ldots & w_{1i} & \ldots & w_{1m} \\ w_{21} & \ldots & w_{2i} & \ldots & w_{2m} \end{bmatrix}^T, \mu = \begin{bmatrix} \mu_{11}, \cdots, \mu_{1m} \\ \mu_{21}, \cdots, \mu_{2m} \end{bmatrix}^T$.

Each adaptive approximation includes an RBFNN and its adaptive laws, and the online-tuning RBFNN is deployed to eliminate the uncertainties in the mechanical dynamics or the hydraulic dynamics. These approximations reduce the chattering effects and improve the precisions. The adaptive laws are derived from the Lyapunov approach [29] and the projection algorithm [31]. The approximation will compensate for the mechanical uncertainties and the hydraulic uncertainties such that

$$D = D^* + \varepsilon = trace\left(W^{*T}\mu\right) + \varepsilon \tag{35}$$

where $\varepsilon = \begin{bmatrix} \varepsilon_1 & \varepsilon_2 \end{bmatrix}^T$ is the reconstructed vector, and $W^*$ is an optimal parameter of $W$, in the RBFNN.

The approximated vector is expressed as the following form:

$$\hat{D} = trace\left(\hat{W}^T\mu\right) \tag{36}$$

where $\hat{W}$ are the estimated parameters of the RBFNN. An approximation error vector $\widetilde{D}$ is defined as follows:

$$\begin{aligned} \widetilde{D} &= D - \hat{D} = D^* + \varepsilon - \hat{D} \\ &= trace\left(W^{*T}\mu\right) - trace\left(\hat{W}^T\mu\right) + \varepsilon = trace\left(\widetilde{W}\mu\right) + \varepsilon \end{aligned} \tag{37}$$

where $\widetilde{W} = W^* - \hat{W}$.

### 3.2.2. Proposed Control

The virtual control (26) is represented as follows:

$$x_{3d} = g_1^{-1}(-\lambda_1 \dot{e}_1 + \dot{x}_{2d} - f_1 - k_1 s_1 - \hat{\delta}_1 - \eta_1 \tanh(\frac{s_1}{\phi_1})) \tag{38}$$

Additionally, the control input (27) is also rewritten as follows:

$$u = g_0^{-1}(\dot{x}_{3d} - f_0 - k_2 s_2 - g_1 s_1 - \lambda_2 e_3 - \hat{\delta}_2 - \eta_2 \tanh(\frac{s_2}{\phi_2})) \tag{39}$$

The uncertainties are estimated by $\hat{d}_i (i = 1, 2)$ in (36), and their weights are adjusted by

$$\dot{\hat{W}}_i = \begin{cases} \lambda_{1i} s_i \mu_i & \begin{aligned} &if\left(\|\hat{W}_i\| < b_{w_i}\right) \\ &or\left(\|\hat{W}_i\| = b_{w_i} \ and \ s_i \hat{W}_i^T \mu_i \geq 0\right) \end{aligned} \\ \lambda_{1i} s_i \mu_i + \frac{\lambda_{1i} s_i \mu_i \hat{W}_i^T \hat{W}_i}{\|\hat{W}_i\|^2} & otherwise \end{cases} \tag{40}$$

where $\|.\|$ denotes the Euclidean norm; $\lambda_i (i = 1, 2)$ are positive learning rates; and $b_W$ is given positive parameter bounds.

In order to prove the stability of the proposed control, the Lyapunov function candidate is redefined as

$$V_2 = \frac{1}{2} \sum_{i=1}^{2} \left( s_i^2 + \frac{1}{2\lambda_i} \widetilde{W}_i^T \widetilde{W}_i \right) \tag{41}$$

**Assumption 2.** *The reconstructed error, $\varepsilon_1$, in the mechanical dynamics and the uncertainties, $\varepsilon_2$, in the hydraulic dynamics are bounded by $\|\varepsilon_i\|_1 < \eta_i (i = 1, 2)$.*

The differential Lyapunov function candidate (41) is expressed as follows:

$$\dot{V} = \sum_{i=1}^{2} \left( s_i \dot{s}_i - \frac{1}{\lambda_i} \dot{\hat{W}}_i^T \widetilde{W}_i \right) = -\zeta^T \Xi \zeta - \sum_{i=1}^{2} \left( \left[ \frac{1}{\lambda_i} \dot{\hat{W}}_i^T - s_i \mu^T \right] \widetilde{W}_i + \left( \varepsilon_i - \eta_i \tanh\left(\frac{s_i}{\phi_i}\right) \right) s_i \right) \tag{42}$$

Let's define $V_{w_i} = \left[ \frac{1}{\lambda_i} \dot{\hat{W}}_i^T - s_i \hat{\mu}^T \right] \widetilde{W}_i (i = 1, 2)$, then (42) can be rewritten as follows:

$$\dot{V} == -\zeta^T \Xi \zeta - \sum_{i=1}^{2} \left( V_{Wi} + \left( \varepsilon_i - \eta_i \tanh\left(\frac{s_i}{\phi_i}\right) \right) s_i \right) \tag{43}$$

By the first equation in (40)

$$V_{wi} = \left[ \frac{1}{\lambda_i} \dot{\hat{W}}_i^T \widetilde{W}_i - s_i \widetilde{W}_i^T \hat{\mu} \right] = 0 \tag{44}$$

By the second equation in (40):

$$V_{w_i} = \lambda_i s_i \hat{\mu}^T \hat{W}_i \hat{W}_i^T \widetilde{W}_i / \|\hat{W}_i\|^2 \tag{45}$$

If the conditions ($\|\hat{W}_i\| = b_w$ and $s_i \hat{W}_i^T \hat{\mu} < 0$) are met, then the condition $\hat{W}_i^T \left( W_i^* - \hat{W}_i \right) = 0.5(\|W_i^*\|^2 - \|\hat{W}_i\|^2 - \|W_i^* - \hat{W}_i\|^2) < 0$ is obtained because of $\|W_i^*\| < b_W$. From (44) and (45), the $V_{W_i}$ is positive.

Based on the analysis as presented above, the derivative Lyapunov function can be represented as follows:

$$\dot{V} = \sum_{i=1}^{2}\left(s_i\dot{s}_i - \frac{1}{\lambda_i}\dot{\hat{W}}_i{}^T\widetilde{W}\right) \leq -\zeta^T\Xi\zeta - \sum_{i=1}^{2}\left(\varepsilon_i - \eta_i\tanh\left(\frac{s_i}{\phi_i}\right)s_i\right) \leq -\mu_{\min}(\Xi)\zeta^T\zeta \qquad (46)$$

where $\mu_{\min}(\Xi)$ is a minimum eigenvalue of the matrix $\Xi$. Because $\dot{V}$ is a negative semi-definite function, $V\left(\zeta(t),\widetilde{W}\right) \leq V\left(\zeta(0),\widetilde{W}\right)$. This means that $s_1, e_3, z_3$ and $\widetilde{W}$ are bounded. Let's define $A_v = -\mu_{\min}(\Xi)\zeta^T\zeta \leq -\dot{V}$ and integrate function $A_V$ in respect to time

$$\int_0^t A_v(t)dt \leq V\left(\zeta(0),\widetilde{W}\right) - V\left(\zeta(t),\widetilde{W}\right) \qquad (47)$$

Since $V\left(\zeta(0),\widetilde{W}\right)$ is a bounded function, and $V\left(\zeta(t),\widetilde{W}\right)$ is a non-increasing and bounded function, the following result can be concluded:

$$\lim_{t\to\infty}\int_0^t A_v(\tau)d\tau < \infty \qquad (48)$$

In addition, as $\dot{A}_V(t)$ is bounded by Barbalat's lemma [25], it can be presented that $\lim_{t\to\infty} A_V(t) = 0$. This means that the vectors $s_1$, $e_3$, and $z_3$ will converge to zero as $t \to \infty$. With this result, the proposed control and adaptive laws guarantee the stability and robustness of the controlled system under the uncertainties.

## 4. Experimental Results

### 4.1. Electrohydraulic System Description

An EEM as shown in Figure 6 is set up to verify the effectiveness of the proposed control scheme. It is an elastic manipulator that contains an adjustable stiffness mechanism and is powered by an electrohydraulic system. In the EHSs, a linear cylinder is used to control the inner link angle. A servo-valve (D633-317B of MOOG company) is used to adjust the flow rate of the cylinder from a hydraulic power unit (Kopack Engineering company). Additionally, an inverter (SINAMICS V20, SIEMENS company) is used to control the flow rate and displacement of the electrohydraulic unit to make differential working conditions. Furthermore, two pressure sensors with a pressure range of $0\text{--}160.10^5$ (N/m$^2$) (Kobold) are used to measure the pressures of the two chambers, and to estimate the feedback torque. The system pressure is limited to $150.10^5$ (N/m$^2$) by a relief valve for safety. Two high-resolution encoder sensors are used to measure the angle of the inner link and the outer link. The adjustable stiffness mechanism includes two springs, a DC motor, and ball screws, which are used to connect the inner link and the outer link. The stiffness of the system is influenced by the angular displacement of the DC motor. So, the desired stiffness can be achieved by controlling the angular displacement of the motor. Another encoder and a motor driver (MD03) are used to carry out this task.

The detailed system parameters of the EEM are given as follows: $J_1 = 0.45$ Nm$^2$, $J_2 = 0.1$ Nm$^2$, $m_1 = 7$ Kg, $m_2 = 3$ Kg, $l_1 = 0.245$ m, $d_1 = 0.215$ m, $d_2 = 0.245$ m, $g = 9.81$ ms$^{-2}$, $P_o = 0\left(\text{N/m}^2\right)$, $k_t = 6.83 \times 10^{-11}$ m$^4/\left(\text{sVN}^{1/2}\right)$, $P_s = 150.10^5\left(\text{N/m}^2\right)$, $\beta = 1.25 \times 10^9$ Nm$^{-2}$, $A_1 = 16.61 \times 10^{-4}$ m$^2$, $A_2 = 3.8 \times 10^{-4}$ m$^2$.

In this paper, all experiments are carried out using MATLAB with the real-time target tool which is supported to embed in two data acquisition (DAQ) cards, (PCI1711 and PCI-Quad04; Measurement Computing and Advance-Tech companies, respectively). The sample time for the signal processing and for implementing the control algorithms was $10^{-3}$ (s).

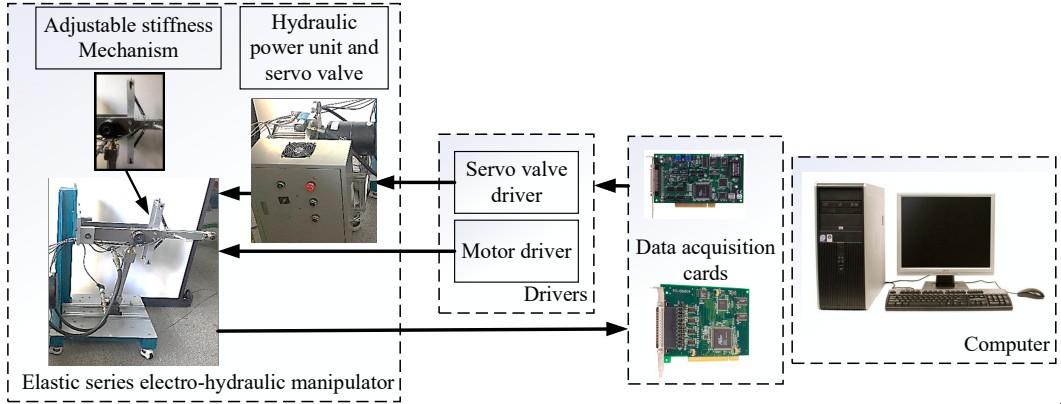

**Figure 6.** Structure of the EEM test bench.

## 4.2. Performance Indexes

In order to validate the quality of each control algorithm, the following performance indexes [32] are used.

(1) $L_2(e) = \sqrt{(1/T_f)\int_0^{T_f} |e|^2 dt}$ is the scalar valued $L_2$ norm and is used as an objective numerical measure of average tracking performance for the entire error curve e (t), where $T_f$ expresses the total running time.

(2) $e_F = \max_{T_f - 2 \le t \le T_f} \{|e(t)|\}$, which is the maximum absolute value of the tracking error during the last two seconds and is used as an index to verify the final tracking accuracy.

## 4.3. The Experimental Procedures

The equilibrium position of the EEM is considered. Two experiments are carried out to verify the effectiveness of the proposed control, the adaptive backstepping sliding mode control (ABSMC). The system carried a low load of 50N, and the stiffness was changed from low stiffness to high stiffness at the 10th second in each experiment.

Firstly, the desired equilibrium position is defined as $x_{1d} = 12 + 10\cos(\pi t)$. The initial state of the electrohydraulic elastic manipulator is $x(0) = \begin{bmatrix} 0 & 0 \end{bmatrix}^T$. To demonstrate the effectiveness of the proposed controller, it is compared to the PID control and the CBSMC.

The parameters of controllers are selected as the PID control: $K_p = 10$, $K_i = 5$, $K_d = 0.1$; CBSMC: $k_1 = 300$, $k_2 = 7.2 \times 10^{10}$, $\lambda_1 = 40$, $\lambda_2 = 4.4$; RBFNN approximations: $E = \begin{bmatrix} e_1, \frac{e_2}{10}, \frac{e_3}{2} \end{bmatrix}^T$ $v_{11} = [1,0,0]^T$, $v_{12} = [0,1,0]^T$, $v_{21} = [1,0,0]^T$, $v_{22} = [0,0,1]^T$, $m_{ei} = \begin{bmatrix} 0.1(j-11) & 0.1(j-11) \end{bmatrix}^T$ $(i = 1,2; j = 1,\ldots,20)$, $\sigma_{ij} = 0.3162$ $(i = 1,2; j = 1,\ldots,20)$, $W_i = [0,\ldots,0]^T \in R^{20}, i = 1,2$.

**Remark 2.** *The control parameters are chosen based on a trial and error method. Because the ABSMC were developed based on the CBSMC, so all control gains of the CBSMC are kept in the ABSMC.*

In this research, the experimental results, including the position, position error, torque, and the control signal, are analyzed. They are presented in Figures 7–10, respectively. In each figure, the responses of the PID control, the CBSMC, and the proposed ABSMC with the approximations in mechanical dynamics and hydraulic dynamics are shown.

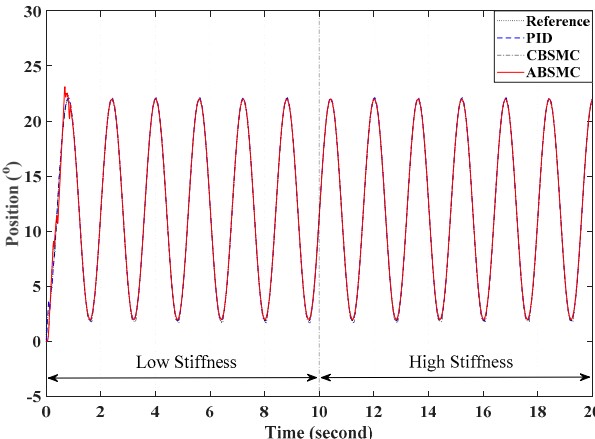

**Figure 7.** The angular displacement responses of the manipulator with the PID control, the CBSMC, and the proposed ABSMC.

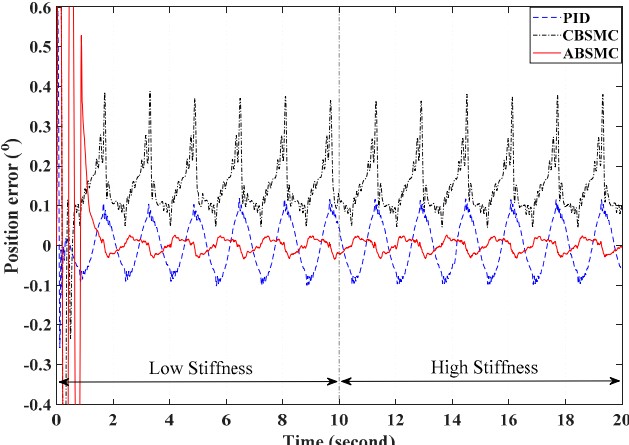

**Figure 8.** The angle error responses of the PID control, the CBSMC, and the proposed ABSMC.

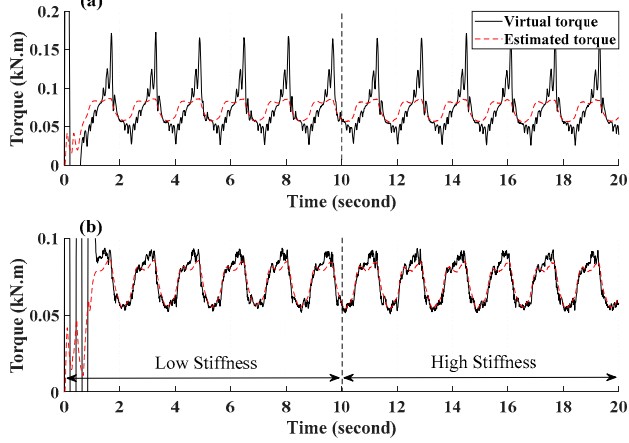

**Figure 9.** The torque responses of (**a**) CBSMC; (**b**) the proposed ABSMC.

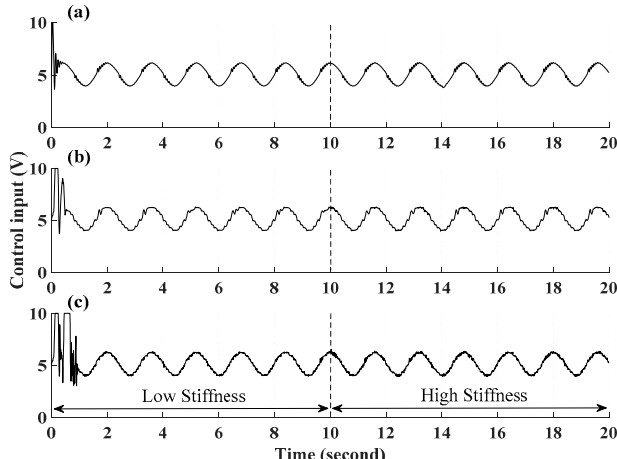

**Figure 10.** The control signal of (**a**) the PID control, (**b**) CBSMC, and (**c**) the proposed ABSMC.

The error responses of PID, CBSMC, and ABSMC are plotted in Figure 8 with the blue dash line, the black dashed dot line, and the red line, respectively. The results exhibited that the ABSMC with approximators improved the precision of the system more effectively than the PID and CBSMC. Figure 9 shows the torque responses of the CBSMC and the ABSMC, with a virtual torque shown as the black line and estimated torque shown as the dashed red line. The results also show that the proposed control improved the tracking torque error by using the approximation to compensate for the uncertainties in hydraulic dynamics. Figure 10 plots the control signals of the PID control, BSMC, and ABSMC. Furthermore, the responses of weights in the approximators are presented in Figure 11. The results verified the convergence of weights of the approximators.

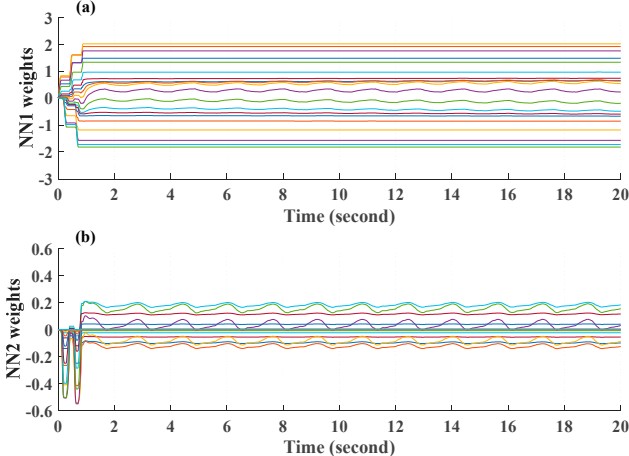

**Figure 11.** The convergence of weights of the approximations (**a**) in mechanical dynamics, (**b**) in hydraulic dynamics.

**Remark 3.** *Because the friction was not considered in the CBSMC, the tracking error of the CBSMC did not converge to zero as shown in Figure 8. It was stable at a nonzero value. The proposed control with the approximation in mechanical dynamics compensated for the friction to improve the accuracy of the controlled system.*

Another experiment is executed to discuss the multi-step tracking ability of the proposed controller. The three controllers are then run for the different positions shown in Figure 12, which has a large distance and a maximum angular displacement; both are near their physical limits. The tracking errors are shown in Figure 13. The results show that the PID control cannot handle such a large movement

well, with the tracking error around 10. The proposed ABSMC has a better tracking performance than the CBSMC with the steady state error quickly converging to zero.

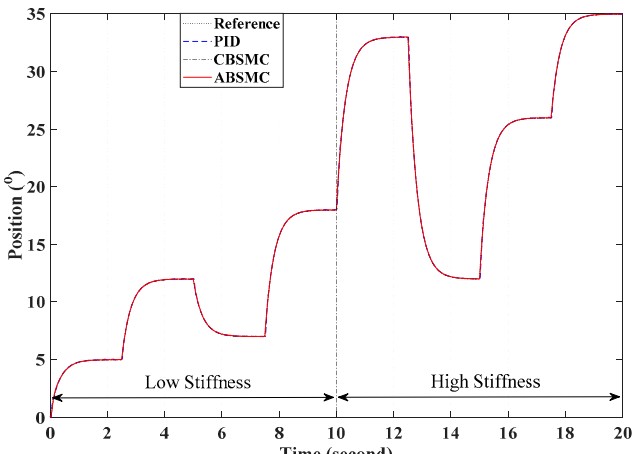

**Figure 12.** The angular displacement responses to the multistep reference of the manipulator with the PID control, the CBSMC, and the proposed ABSMC.

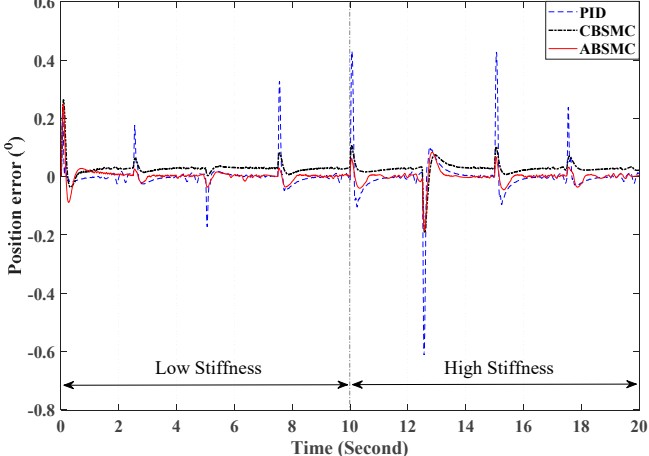

**Figure 13.** The angular error responses to the multistep reference of the manipulator with the PID control, the CBSMC, and the proposed ABSMC.

**Remark 4.** *Because the ABSMC is developed based on the CBSMC, the parameters of the ABSMC will preserve the parameters of the CBSMC. The steady-state errors of the CBSMC still exist as shown in Figures 8 and 13 due to the effect of the friction, gravity, and modeling error. In order to exhibit the effectiveness of the proposed control, the friction, gravity, and modeling error are substantially considered in the CBSMC. The ABSMC with the RBFNNs compensated for these uncertainties to improve the steady-state errors as exhibited in Figures 8 and 13.*

**Remark 5.** *The parameters of the PID control are chosen by a trial and error method to get optimal values. Its performances are compared with the proposed control to show the effectiveness of the proposed control.*

**Remark 6.** *In Figures 8–10, the result showed that the chattering effect occurred due to the input constraints of the actuator system. As shown in (46), the control gains will affect the time to convergence of the tracking errors. When the control gains and the position errors are significant, the control signal will be significant and the input constraint will happen. This issue caused the chattering phenomenon. The solution to this issue will be discussed in future work.*

The calculated performance indexes [32] from the experimental results are shown in Table 1. The results proved the effectiveness of the proposed control in improving the tracking error of the controlled system.

**Table 1.** Performance Indexes.

| | Controller | PID | CBSMC | ABSMC |
|---|---|---|---|---|
| Sin | Index 1 | 0.2162 | 0.5479 | 0.0614 |
| | Index 2 | 0.1811 | 0.5673 | 0.0548 |
| Multistep | Index 1 | 0.4083 | 0.1419 | 0.1083 |
| | Index 2 | 1.3697 | 0.3248 | 0.3145 |

## 5. Conclusions

In this paper, an adaptive backstepping control based RBFNN is proposed for equilibrium position control of the EEM with the presence of the variant stiffness and the uncertainties. The uncertainties are matched and unmatched uncertainties because the system dynamics considered the actuator dynamics. By using the sliding mode control and RBFNN with the backstepping technique, the proposed control compensated for both the unknown matched and unmatched uncertainties in the mechanical dynamics and actuator dynamics to improve the control performance. Additionally, the stability of the controlled system is theoretically proven by the Lyapunov approach and backstepping technique. Finally, some experiments were conducted, and the results were compared to the PID and CBSMC to show the effectiveness of the proposed control with the variant stiffness. In future work, the stiffness controller and the equilibrium position control will be implemented at the same time to exhibit the ability of the EEM.

**Author Contributions:** K.K.A. was the supervisor providing funding and administrating the project, and he reviewed and edited the manuscript. D.-T.T. did the investigation, methodology, analysis, and the validation, wrote the software, and wrote the original draft. M.-N.N. did experiments and analyzed the results.

**Funding:** This work was supported by the Basic Science Research Program through the National Research Foundation of Korea (NRF) funded by the Korean government (MEST) (NRF-2017R1A2B3004625).

**Conflicts of Interest:** The author declares no conflict of interest.

## Appendix A

According to Figure 2 and (7), the dynamic equations of a two-link robotic manipulator can be expressed as [27]

$$M(\theta)\ddot{\theta} + C(\theta, \dot{\theta}) + G(\theta) = \tau + \tau_f \tag{A1}$$

with $M(\theta) = \begin{bmatrix} J_1 + J_2 + m_2 l_1^2 + 2m_2 l_1 d_2 \cos\theta_2 & J_2 + m_2 l_1 d_2 \cos\theta_2 \\ J_2 + m_2 l_1 d_2 \cos\theta_2 & J_2 \end{bmatrix}$, $C(\theta, \dot{\theta}) = \begin{bmatrix} -m_2 l_1 d_2 \sin\theta_2(\dot{\theta}_2^2 + 2\dot{\theta}_1\dot{\theta}_2) \\ m_2 l_1 d_2 \sin\theta_2\dot{\theta}_1^2 \end{bmatrix}$, $G(\theta) = \begin{bmatrix} g(m_1 d_1 + m_2 l_1)\cos\theta_1 + gm_2 d_2 \cos(\theta_1 + \theta_2) \\ gm_2 d_2 \cos(\theta_1 + \theta_2) \end{bmatrix}$, $\theta = \begin{bmatrix} \theta_1 & \theta_2 \end{bmatrix}^T$, $\dot{\theta} = \begin{bmatrix} \dot{\theta}_1 & \dot{\theta}_2 \end{bmatrix}^T$, $\ddot{\theta} = \begin{bmatrix} \ddot{\theta}_1 & \ddot{\theta}_2 \end{bmatrix}^T$, $\tau = \begin{bmatrix} \tau_1 & \tau_2 \end{bmatrix}^T$ where $J_i$ are inertia moments about an axis through the mass center of the *ith* link; $m_i$ are the weights of the *i*th link; $d_i$ are the distance from the center of an *i*th joint to the mass center of the *i*th link; $\ddot{\theta}$ and $\dot{\theta}$ are the acceleration vector and the velocity vector; $g$ is the acceleration of gravity; and $\tau_i$ are the torque inputs generated by *i*th joint.

## Appendix B

From (16), the state space of the EHSs is shown as:

$$
\begin{aligned}
\dot{x}_1 &= x_2 \\
\dot{x}_2 &= g_1 x_3 + f_1 + \delta_1(t) \\
\dot{x}_3 &= g_0 u + f_0 + \delta_2(t)
\end{aligned}
\tag{A2}
$$

where $g_1 = \dfrac{J_2 l_1}{J_1 J_2 + J_2 l_1^2 m_2 - (d_2 l_1 m_2 c_2)^2}$, $f_1 = -\Bigg(\dfrac{(J_2 + d_2 l_1 m_2 c_2)\Big(d_2 l_1 m_2 s_2 \dot{\theta}_1^2 + d_2 g m_2 c_{12}\Big)}{J_1 J_2 + J_2 l_1^2 m_2 - (d_2 l_1 m_2 c_2)^2}$

$+ \dfrac{J_2\big(d_2 \dot{\theta}_2 l_1 m_2 s_2 (\dot{\theta}_2 + 2\dot{\theta}_1) - g c_2 (d_1 m_1 + l_1 m_2) - d_2 g m_2 c_{12}\big)}{J_1 J_2 + J_2 l_1^2 m_2 - (d_2 l_1 m_2 c_2)^2}\Bigg)$, $g_0 = \beta C_d \omega k_s \sqrt{\dfrac{2}{\rho}}\Big(\dfrac{A_1 \sqrt{\Delta_1}}{V_{01} + A_1 x_c} + \dfrac{A_2 \sqrt{\Delta_2}}{V_{02} - A_2 x_c}\Big)$, $f_0 = $

$-\beta x_2\Big(\dfrac{A_1^2}{V_{10} + A_1 x_c} + \dfrac{A_2^2}{V_{20} - A_2 x_c}\Big)$, with $c_i = \cos(\theta_i)$, $s_i = \sin(\theta_i)$ $(i = 1, 2)$, and $c_{12} = \cos(\theta_1 + \theta_2)$.

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
