# Peer review of "RBF Neural Network Based Backstepping Control for an Electrohydraulic Elastic Manipulator"

_applsci, doi:10.3390/app9112237_

Round 1
Reviewer 1 Report
Dear Authors
Major criticism:
Figure 1 needs improvement. Symbols do not conform to ISO 1219-1 and ISO 1219-2. (mainly actuator symbol and valve symbol). The pressure should be given in SI units and not in the Bar.
The proposed changes in the pdf file.

Author Response
Please see an attached file.

Reviewer 2 Report
good idea, very useful
Author Response
Please see an attached file.

Reviewer 3 Report
In this work, the authors investigated the control of an electro-hydraulic elastic manipulator using RBF neural network based backstepping control. Both theoretical analysis and experimental verification are provided. Overall, this paper can be accepted after a minor revision considering the following comments:
1. Non-smooth Right Hand Side. In the control law, to reject the unknown disturbance, a sliding mode control with the presence of sign function is employed in the expression. Note that for sign function, Lipschitz Condition does not hold and Clake's general derivative usually is necessary for rigorous analysis to conclude the theoretical results on convergence. I suggest the authors to update the proof accordingly. Otherwise, the conventional Lyapunov analysis which requires Lipschitz condition to hold may not apply (Eqs. (28) to (30)).
2. Generalization. In principle, the investigated system can be regarded as a special type of SEA (series elastic actuator). In this sense, is it possible to generalize the current control scheme to a broad class of SEA systems? If yes, how?
3. Neural Structure. In the structure of the proposed RBF neural network, it seems there is no cross linking between layer 2 and layer 3 for delta_1 and delta_2? A general neural architecture usually has all to all connection between neighboring layers. Please explain why do you particularly select this structure and preferably compare in simulation/experiment to show the advantage of this scheme.
4. Approximation Error. It is assumed that the RBF network is able to represent the uncertainty. However, in practice, there may be a residual approximation error due to the use of limited number of neurons in hidden layers. Accordingly, there will be an un-compensated term appearing on the right hand side of the control equation. Please discuss on this.
5. Fluctuation of Estimated Neural Weights. As observed in Fig. 11, the neural weights fluctuate with time. Intuitively, given enough time, the neural weights should be able to converge to the true value, which is a constant. This may contradict the observation from Fig. 11. Please explain.
6. Practical Consideration. For the experimental system, how large is the sensing noise? Note that the sensing noise might be totally random which cannot be estimated using a model. In this sense, the RBF neural model may fail to estimate or compensate its value. Also, due to the fact that the manipulator has a hydraulic actuation mechanism, which usually has a low bandwidth, and suffers from a non-ignorable time delay. In the presented control scheme, it seems that there is no control term particularly to compensate the delay and low bandwidth. If that is true, this might result in some imperfection in the experimental results. Could you please explain which part in the experiment accounts for this?
7. The following papers are related:
[A] M. Yue, L. Wang, and T. Ma, “Neural network based terminal sliding mode control for WMRs affected by an augmented ground friction with slippage effect,” IEEE/CAA Journal of Automatica Sinica, 4(3), 498-506, 2017
[B] H. J. Yang and J. K. Liu, “An adaptive RBF neural network control method for a class of nonlinear systems,” IEEE/CAA J. of Autom. Sinica, vol. 5, no. 2, pp. 457-462, Mar. 2018.
[C] S. Li, M. Zhou, and X. Luo, “Modified Primal-Dual Neural Networks for Motion Control of Redundant Manipulators with Dynamic Rejection of Harmonic Noises,” IEEE Transactions on Neural Networks and Learning Systems, 29(10), pp. 4791 - 4801, Oct. 2018.
Overall, this work presents a model with RBF neural networks to approximate the uncertainty, using backstepping to deal with the under-actuation, and sliding mode control to achieve a robust controller for electro-hydraulic elastic manipulator. Both theoretical analysis and experimental verification are conducted. I suggest the acceptance of this work provided the above comments are full addressed.
Author Response
Please see an attached file.
